# Analysis of Brain, Blood, and Testis Phenotypes Lacking the *Vps13a* Gene in C57BL/6N Mice

**DOI:** 10.3390/ijms25147776

**Published:** 2024-07-16

**Authors:** Jitrapa Pinyomahakul, Masataka Ise, Meiko Kawamura, Takashi Yamada, Kentaro Okuyama, Shinsuke Shibata, Jun Takizawa, Manabu Abe, Kenji Sakimura, Hirohide Takebayashi

**Affiliations:** 1Division of Neurobiology and Anatomy, Graduate School of Medical and Dental Sciences, Niigata University, Niigata 951-8510, Japan; j.pinyomahakul@gmail.com (J.P.);; 2Department of Animal Model Development, Brain Research Institute, Niigata University, Niigata 951-8585, Japan; meiko@bri.niigata-u.ac.jp (M.K.); manabu@bri.niigata-u.ac.jp (M.A.); sakimura@bri.niigata-u.ac.jp (K.S.); 3Department of Hematology, Endocrinology and Metabolism, Faculty of Medicine, Niigata University, Niigata 951-8510, Japan; takayamada@hotmail.com (T.Y.); juntaki@med.niigata-u.ac.jp (J.T.); 4Division of Microscopic Anatomy, Graduate School of Medical and Dental Sciences, Niigata University, Niigata 951-8510, Japan; k2okuyama@med.niigata-u.ac.jp (K.O.); shibatas@med.niigata-u.ac.jp (S.S.); 5Center for Coordination of Research Facilities, Niigata University, Niigata 951-8510, Japan

**Keywords:** chorea-acanthocytosis (ChAc), knockout (KO) mice, VPS13A (vascular protein sorting 13A)/chorein, conditional allele, lipid droplet, mitochondria-associated endoplasmic reticulum membranes (MAMs)

## Abstract

The *Vps13a* gene encodes a lipid transfer protein called VPS13A, or chorein, associated with mitochondria-associated endoplasmic reticulum (ER) membranes (MAMs), mitochondria–endosomes, and lipid droplets. This protein plays a crucial role in inter-organelle communication and lipid transport. Mutations in the *VPS13A* gene are implicated in the pathogenesis of chorea-acanthocytosis (ChAc), a rare autosomal recessive neurodegenerative disorder characterized by chorea, orofacial dyskinesias, hyperkinetic movements, seizures, cognitive impairment, and acanthocytosis. Previous mouse models of ChAc have shown variable disease phenotypes depending on the genetic background. In this study, we report the generation of a *Vps13a* flox allele in a pure C57BL/6N mouse background and the subsequent creation of *Vps13a* knockout (KO) mice via Cre-recombination. Our *Vps13a* KO mice exhibited increased reticulocytes but not acanthocytes in peripheral blood smears. Additionally, there were no significant differences in the GFAP- and Iba1-positive cells in the striatum, the basal ganglia of the central nervous system. Interestingly, we observed abnormal spermatogenesis leading to male infertility. These findings indicate that *Vps13a* KO mice are valuable models for studying male infertility and some hematological aspects of ChAc.

## 1. Introduction

Chorea-acanthocytosis (ChAc) is a group of neuroacanthocytosis syndromes predominantly manifested by chorea and orofacial dyskinesias [1,2]. Mutations in the vacuolar protein sorting 13A (*VPS13A*) gene, encoding the protein VPS13A (also known as chorein), have been reported to be responsible for the pathogenesis of ChAc (OMIM:200150) in humans [3,4,5]. These syndromes are rare autosomal recessive inherited disorders characterized by progressive neurodegeneration and abnormally shaped erythrocytes with spikes, known as acanthocytes, with onset in the third to fifth decades of life. A patient with ChAc exhibits hyperkinetic movements, seizures, cognitive impairment, neuropsychiatric symptoms, and acanthocyte detection in peripheral blood smears.

Previous studies have shown that the VPS13A protein is found in close proximity to the membrane contact sites between the mitochondria–endoplasmic reticulum (ER), known as mitochondria-associated ER membranes (MAMs), and mitochondria–endosomes. In addition, the VPS13A protein is also present at the contact sites of lipid droplets [6,7]. The full range of functions for the VPS13A protein is still being actively investigated; some research suggests it plays crucial roles in inter-organelle communication [8] and lipid transport [6].

The first conventional *Vps13a* knockout (KO) mice were generated and developed for a mouse model of ChAc [9]. Gene-targeting techniques were used to delete exons 60–61 of the mouse *Vps13a* gene in embryonic stem (ES) cells (CCE, 129/Sv background). Although these mice showed acanthocytosis and motor disturbance in old age, the resulting phenotype was relatively mild. It has been reported that the genetic background of the ChAc mouse models affected the phenotypic severity [10]. They reported that *Vps13a*-mutant mice on the 129S6/Sv and FVB backgrounds displayed significantly aberrant motor function and behavior, while no marked irregularities were observed on the C57BL/6J background. Additionally, male ChAc-model mice were found to be completely infertile due to significantly reduced sperm motility, probably due to mitochondrial abnormalities in the midpiece of the sperm, which is a VPS13A-rich region [11].

The purpose of this study was to generate novel *Vps13a* flox (*Vps13a^flox^*) and *Vps13a* null (*Vps13a^-^*) alleles in a pure C57BL/6N mouse background. In addition, we generated *Vps13a* KO mice and investigated brain, blood, and testis phenotypes by histological analyses.

## 2. Results

### 2.1. Establishment of a Novel Vps13a Mutant Line on the C57BL/6N Background

To generate *Vps13a* null allele, we first generated the *Vps13a^flox^* allele in which loxP sequences were inserted to delete exons 60 and 61 in the ES cells of the C57BL/6N background (Figure 1A). After generating chimera mice, heterozygous *Vps13a^flox^+neo* mice were obtained by crossing C57BL/6N mice. Then, the *Vps13a^flox^+neo* mice were crossed with *Actb-iCre* mice [12] to generate the *Vps13a* null (*Vps13a^-^*) allele. *Vps13a* KO mice were generated by mating *Vps13a^-^* heterozygous mice. The genotyping PCR analysis confirmed that the observed DNA size corresponded to the expected Wt genotype at 335 bp and *Vps13a* KO at 686 bp (Figure 1B). The findings of the Western blot analysis revealed the absence of VPS13A protein bands at 360 kDa in *Vps13a* KO mice (Figure 1C).

### 2.2. Neurological Phenotype of Vps13a KO Mice

Initially, the gross phenotype of the *Vps13a* KO mice looked normal until old age, at more than one and a half years. Because the previously reported *Vps13a* KO mice exhibited multiple apoptotic cells and increased GFAP-positive cells in the striatum, together with a mild neurological phenotype [9], we examined the histological phenotype of *Vps13a*-mutant mice at 3 months old (3MO) and 18 months old (18MO). Nissl staining of the *Vps13a* KO mouse brains revealed no significant differences between the Wt and *Vps13a* KO mice at 3MO (Figure 2A) and 18MO (Figure 3A).

We also examined glial activation by GFAP and Iba1 IHC. The results indicate no apparent difference between Wt and *Vps13a* KO striatum in the 3MO and 18MO mice (Figure 2B–E and Figure 3B–E). We conducted Sox9 IHC staining as a pan-astrocyte marker [13] to verify the number of astrocytes in the striata of *Vps13a*-mutant mice at 3MO (Appendix A). No disparity was seen in the number of astrocytes between the Wt and KO brains. Finally, we performed *D1R* and *D2R* in situ hybridization (ISH) in the 18MO *Vps13a* KO striatum (Appendix A) to observe medium spiny neurons of direct or indirect pathways in the striatum. We did not see any significant differences between the Wt and KO mice in terms of *D1R-* and *D2R*-positive cells.

### 2.3. Transmission Electron Microscopy Analysis of Sperm

The absence of the VPS13A protein has been reported to cause mitochondrial abnormality at the midpiece of sperm in *Vps13a* KO mice [11]. Since we also observed male infertility in the *Vps13a* KO mice on the C57BL/6N background, we examined sperm using transmission electron microscopy (TEM). We observed a marked morphological difference between the *Vps13a* KO mice and the control mice (*Vps13a* Het) at 4 months old. Most of the sperm from the *Vps13a* KO mice exhibited an aberrant mitochondrial morphology in the midpiece region (Figure 4). As an abnormal morphology of mitochondria, we observed mitochondria with onion-like structures characterized by concentric layers of cristae membranes and spotted mitochondria. In addition to the irregularity of the cristae, we also found that the mitochondrial matrix exhibited a relatively low electron density in the *Vps13a* KO mice (Figure 4).

### 2.4. Histological Analyses of Male Reproductive Organs

The abnormal sperm morphology led us to examine male reproductive organs at a later stage. The H&E staining revealed dilated lumens in seminiferous tubules of the *Vps13a* KO testis at 10 months old (Figure 5A, Ctrl: 12.10 ± 0.34, *n* = 4 mice, KO: 24.93 ± 1.60, *n* = 3 mice, *p* = 0.0003, unpaired *t*-test). Furthermore, the epididymal ducts of these *Vps13a* KO mice exhibited an absence of spermatozoa (Figure 5B). We also examined the mRNA expression of the *Vps13* family (*Vps13a*, *Vps13b*, *Vps13c*, and *Vps13d*) in control and *Vps13a* KO testes. Among them, *Vps13a* mRNA was highly expressed in the spermatocytes in the seminiferous tubules in both the control and *Vps13a* KO mice (Appendix A).

### 2.5. Analyses of Blood Phenotype

To examine the peripheral blood phenotype of the *Vps13a* KO mice, we performed smear tests using New Methylene Blue staining followed by May-Grunwald staining (Figure 6A). Although we did not observe acanthocytes in the smear tests, we observed an increased percentage of reticulocytes in the peripheral blood of *Vps13a* KO mice (Figure 6B), suggesting increased red blood cell production in the bone marrow.

## 3. Discussion

This study reports the generation of novel *Vps13a^flox^* and recombined null alleles in the C57BL/6N background. We observed male infertility, abnormal sperm morphology, and increased reticulocytes in peripheral blood from the *Vps13a* KO mice; however, there were no significant abnormalities in the forebrain of KO mice. Although conventional *Vps13a* KO mice have been reported previously, this is the first report of the *Vps13a* flox allele in the C57BL/6N background, and this allele is a valuable resource for conditional analysis, especially in the testis and blood systems.

In humans, *VPS13A* gene mutations cause ChAc, which is associated with various neurological symptoms in the central nervous system and histological changes in the striatum [14]. In a previous report, brain phenotypes such as increased GFAP staining were observed in the *Vps13a* KO striatum, and backcrossing to the C57BL/6J strain weakened the phenotype [10]. The neural phenotypes were almost intact in the *Vps13a* KO mice in the C57BL/6N background generated in this study. In other words, the deletion of the *Vps13a* gene in mice differs among mouse strains, and a weak phenotype was observed in the C57BL/6 mice, suggesting the presence of a modifier gene in the mouse genome. Although *Vps13a* KO brains in the C57BL/6 strain were almost normal under specific-pathogen-free laboratory conditions, the phenotypic severity may be altered by environmental factors such as stress, infection, and food.

The *Vps13a* KO mice in the C57BL/6 strain exhibited male infertility and abnormal sperm morphology, consistent with previous reports on *Vps13a* KO mice in other mouse strains [10]. A previous study indicated that a *Vps13a* deficiency causes male infertility characterized by reduced sperm motility, probably due to abnormal mitochondrial function in the sperm midpiece [11]. We also observed abnormal mitochondrial morphology in the midpiece of sperm by TEM analysis. A recent study further reported that mitochondrial ultrastructural abnormalities were observed specifically during the late stages of sperm maturation [15]. There are four members of the *Vps13* family—*Vps13a*, *Vps13b*, *Vps13c*, and *Vps13d*—and mutations in each gene cause distinct neurological diseases [16]. The ISH results suggest that *Vps13a* is the most strongly expressed member in the testis and has a nonredundant function in male germ cells. *Vps13a* is conserved from yeast to mammalian species. VPS13A forms mitochondrial–ER contact sites (MERCs), which are responsible for phospholipid transport, by binding to endoplasmic-reticulum-localized VAMP-associated protein A (VAPA) [17]. VPS13A also localizes to lipid droplets, which affects their motility [18]. It is interesting to study which function of VPS13A protein contributes to normal spermatogenesis.

In blood smear tests, an increased percentage of reticulocytes in the *Vps13a* KO mice suggests increased red blood cell production in the bone marrow. The osmotic and mechanical fragilities of red blood cells due to *Vps13a* deficiency may lead to the slightly increased destruction of the cells. Although mature red blood cells usually do not have mitochondria, it is known that they retain mitochondria or mitochondrial components in some disease states [19,20]. It is also interesting to study the function of VPS13A in maintaining red blood cell homeostasis [21].

## 4. Materials and Methods

### 4.1. Mice

A bacterial artificial chromosome (BAC) clone containing the *Vps13A* gene (RP23-39A16) was used to construct a targeting vector. The 5′ homology arm of 7.16 kb and 3′ homology arm of 3.06 kb were retrieved from the BAC clone and inserted into the 5′ entry clone (pD5UE-2) and the 3′ entry clone (pD3DE-2), respectively, using the Quick and Easy BAC modification kit (Gene Bridges, Dresden, Germany). A 2.44 kb DNA fragment carrying *Vps13a* exons 60 and 61 was amplified by PCR and subcloned into BclI/SacI sites of a modified middle entry clone (pDME-1; a DNA fragment of loxP sequence and *pgk-Neo* cassette flanked by two FLP recognition target (FRT) sites were sequentially located at the site 320 bp upstream of exon 60, while the other loxP sequence was placed at the site 260 bp downstream of exon 61). For targeting vector assembly, the three entry clones were recombined to a destination vector plasmid (pDEST-DT containing a cytomegalovirus enhancer/chicken beta-actin (CAG) promoter-driven diphtheria toxin gene) using the MultiSite Gateway Three-fragment Vector Construction Kit (Invitrogen; Thermo Fisher Scientific, Inc., Waltham, MA, USA). The targeting vector linearized with NotI was electroporated into the mouse ES cell line, RENKA, derived from the C57BL/6N strain [22]. The culturing of the ES cells and the generation of chimeric mice (*Vps13a*^flox^ mice) were performed as described previously [22]. Homologous recombinant ES clones were identified by Southern blot analysis. The EcoRV-digested genomic DNA was hybridized with a 5′ probe, 12.7 kb for the Wt and 10.8 kb for the targeted allele; the KpnI-digested genomic DNA was hybridized with a 3′ probe, 29.2 kb for the Wt and 8.9 kb for the targeted allele.

All mice had ad libitum access to pelleted food and water. The following measures were taken to minimize the animals’ suffering during experiments: restlessness, vocalizing, loss of mobility, failure to groom, open sores/necrotic skin lesions, guarding (including licking and biting) a painful area, and a change in body color. No mice exhibited these signs during this study, so none were excluded. The study was not preregistered.

### 4.2. Genotyping PCR

Genotyping by PCR was performed as follows: Genomic DNA was extracted from the tail tips or fingers of mice [12,23]. The extracted DNA was used as a template for the PCR reaction using the ΔNeo29810F (5′-TGT CAG TGT CCT GTT ACC TTT C-3′) and flox32794R (5′-GGA TGA GTC AAT ATG TTG TTC GT-3′) as the null primer set for the *Vps13a* null allele, and ΔNeo29810F (5′-TGT CAG TGT CCT GTT ACC TTT C-3′) and ΔNeo30144R (5′-TCC TCT GTG GCT ATT TGC TTC-3′) as the ΔNeo primer set for the Wt allele.

The PCR procedures were started with an initial denaturing at 95 °C for 1 min, followed by denaturing at 95 °C for 20 s, annealing at 58 °C for 30 s, and extension at 72 °C for 30 s, repeating these steps for 32 cycles, and followed by the final extension step at 72 °C for 2 min. Gel electrophoresis was performed on 2% agarose gel containing ethidium bromide in Tris-Acetate-EDTA (TAE) buffer to visualize the PCR products.

### 4.3. Western Blot Analysis

Western blotting was performed as previously described [23]. The frozen mouse forebrain tissues were homogenized in ice-cold homogenization buffer (0.32 M sucrose, 5 mM ethylenediaminetetraacetic acid (EDTA) (pH 8.0), 10 mM Tris-hydrochloride (HCl) (pH 7.4), and phosphatase and protease inhibitors) and centrifuged at 4500 rpm for 10 min at 4 °C. Supernatants were obtained, and the protein concentration was determined using TaKaRa Bicinchoninic Acid (BCA) Protein Assay Kit (#T9300A, TaKaRa, Shiga, Japan). Lysates were resolved by sodium dodecyl sulfate-polyacrylamide gel electrophoresis (SDS-PAGE) and transferred to nitrocellulose membranes. Membranes were blocked with 3% skimmed milk and probed with the following primary antibodies: rabbit polyclonal anti-VPS13A antibody (1:500, #28618-1-AP, Proteintech, Rosemont, IL, USA) and mouse monoclonal anti-glyceraldehyde-3-phosphate dehydrogenase (GAPDH) antibody (1:10,000, #016-25523, Wako, Osaka, Japan). After incubation, the membrane was washed and incubated with horseradish peroxidase (HRP)-conjugated secondary antibodies. Immunoreactive bands were visualized using enhanced chemiluminescence (ECL) and quantified by Fiji (ImageJ2 version 2.14.0/1.54f, National Institutes of Health, Bethesda, MD, USA).

### 4.4. Histology

The mice were anesthetized with an intraperitoneal injection of pentobarbital, a widely used anesthetic agent, and then perfused with 4% paraformaldehyde (PFA) in 0.1 M phosphate-buffered (PB) solution (pH 7.4). The brain and testis were removed and postfixed with the 4% PFA/PB solution. The tissues were dehydrated in a graded ethanol series and embedded in paraffin. Serial sections were cut from the paraffin block. Nissl staining and hematoxylin and eosin (H&E) staining were performed as previously described [24,25].

Immunostaining was performed on paraffin sections (10 μm thick) as previously described [26] using the following antibodies: mouse monoclonal antibodies against rabbit antibodies against glial fibrillary acidic protein (GFAP, 1:10, #422251, Nichirei, Tokyo, Japan); Sox9 (1:400, #33636, Cell Signaling Technology, Danvers, MA, USA); and Iba1 (1:500, 019-19741, Wako, Osaka, Japan). Immunolabeling was detected using an HRP-conjugated anti-rabbit IgG antibody (#458, 1:200; MBL, Nagoya, Japan) and visualized with diaminobenzidine (DAB)/H_2_O_2_ solution. Sections were counterstained with hematoxylin.

*In situ* hybridization (ISH) was performed on paraffin sections (10 μm thick) as previously described [26,27], using a mouse *D1R* (also known as *Drd1*; 1035–3074nt, NM_01176), *D2R* (also known as *Drd2*; 38–1549nt, NM_010077), *Vps13a* (8159–9435nt, NM_173028), *Vps13b* (200–1539nt, NM_177151), *Vps13c* (10579–11542nt, NM_177184), and *Vps13d* (12454–13677nt, NM_001276502) probes. After color development using nitro blue tetrazolium (NBT) and 5-bromo-4-chloro-3-indolyl phosphate (BCIP) solution, sections were counterstained with nuclear fast red staining.

Images were taken using an Olympus microscope (BX53) and digital camera (DP74; Olympus, Tokyo, Japan). ImageJ2 version 2.14.0/1.54f was used to quantify the positive cells on the staining section from the acquired images.

### 4.5. TEM Analysis

The TEM analysis was performed as previously described with slight modifications [28,29]. Briefly, epididymis from each mouse was fixed with 2.5% glutaraldehyde (Nisshin EM, Tokyo, Japan) in 0.1 M cacodylate buffer (pH 7.4) for 24 h at 4 °C and postfixed with 1% OsO4 (Merck, Darmstadt, Germany) for 2 h at 4 °C. The samples were dehydrated through an ascending ethanol series (50, 70, 80, 90, 95, and 100%) and acetone and n-butyl glycidyl ether (QY-1, Okenshoji Co., Ltd., Tokyo, Japan) twice for each, a graded concentration of Epon with QY-1, and embedded in 100% Epon. After polymerization for 72 h at 65 °C, ultrathin sections of 70 nm thickness were cut with an ultramicrotome (UC7, Leica Microsystems GmbH, Wetzlar, Germany). The sections were stained with uranyl acetate and lead citrate for 10 min each and were observed with a transmission electron microscope (H-7650, Hitachi High-Tech, Tokyo, Japan).

### 4.6. Measurement of the Lumen Area in Seminiferous Tubules

Images of the testis sections were obtained from the control (WT and *Vps13a* Het mice) and *Vps13a* KO mice. Oval shapes were drawn over them in the images to measure the radius of the seminiferous tubules and their lumens in the images, as previously described [30]. For each sample, 20–25 seminiferous tubules and their corresponding lumens were measured. The area of the seminiferous tubules and their lumens were then calculated using the oval area formula. To obtain the data as the percentage of the lumen area per the total area of the seminiferous tubules, the lumen area was divided by the area of the seminiferous tubules and multiplied by 100. The mean percentage of the lumen area per the total area of the seminiferous tubule for each sample was calculated. These mean values were then pooled to form the data for the control and *Vps13a* KO groups, and the groups were compared using an unpaired *t*-test.

### 4.7. Blood Smear Test

The reticulocyte assay was performed following the standard method of the International Council for Standardization in Haematology [31] in the Health Laboratory of World Health Organization. To perform this assay, equal volumes of whole blood and New Methylene Blue staining solution (50 µL each) were mixed and incubated at 37 °C for 10 min. Then, the wedge smear technique was used to form the thin blood film on a glass slide. Subsequently, May-Grunwald staining was applied to prevent fading. The reticulocytes were counted among 30,000 RBCs to ensure accuracy. Reticulocytes were identified as cells containing two or more distinct blue granular or reticular staining patterns.

### 4.8. Statistical Analysis

Sample sizes were determined based on previously published papers [12,29]. We report the data as the means ± standard error of the mean (SEM). In histological analyses, a significance assessment was conducted using an unpaired *t*-test. A *p*-value below 0.05 was considered to have statistical significance. GraphPad Prism 9 (GraphPad Software, Inc., San Diego, CA, USA) was used as the statistical data analysis software.

## 5. Conclusions

In this study, we generated novel *Vps13a^flox^* and recombined null alleles with a C57BL/6N background. The *Vps13a* KO mice exhibited male infertility, abnormal sperm morphology, and increased reticulocytes in blood smear tests. Our data indicate that *Vps13a* KO mice are valuable models for studying male infertility and some hematological aspects of ChAc. The *Vps13a^flox^* allele is also useful for studying the cell-autonomous mechanism of the testis and blood phenotypes by conditional KO experiments.

## Figures and Tables

**Figure 1 ijms-25-07776-f001:**
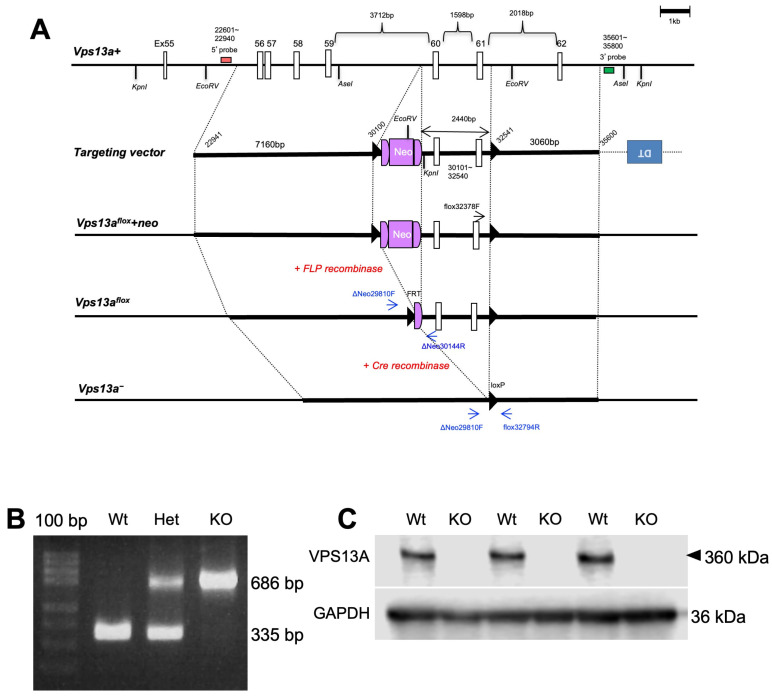
Generation of *Vps13a* knockout (KO) mouse line. (**A**) Schematic targeting strategy of the *Vps13a*-mutant mouse illustrates the wild-type *Vps13a* allele (*Vps13a^+^*), targeting vector, targeted allele after homologous recombination (*Vps13a^flox^+neo*), targeted allele pre-Cre recombination (*Vps13a^flox^*), and the final *Vps13a* knockout allele post-Cre recombination (*Vps13a^-^*). Exons are depicted as white boxes, the neomycin resistance cassette (Neo) as a light purple box, and the diphtheria toxin cassette (DT) as a blue box. The loxP sites are indicated by black triangles. The FRT sites are located at both ends of the Neo cassette. Small red and green boxes indicate 5′ and 3′ probe regions used for Southern blot analysis, respectively. (**B**) PCR analysis confirms the genotypes of wild-type (Wt) mouse by showing a single band at 335 base pairs (bp), heterozygous (Het) mouse as two bands at 335 and 686 bp, and KO mouse as a single band at 686 bp. A 100 bp DNA ladder was used as a size marker. (**C**) Western blot analysis of forebrain homogenates from 4-week-old mice using anti-VPS13A antibody exhibiting the absence of the VPS13A protein in KO mice (*n* = 3/genotype). An arrowhead indicates the predicted size of the VPS13A protein (360 kDa). The Western blot membrane was stripped and re-probed with anti-GAPDH antibody as a loading control.

**Figure 2 ijms-25-07776-f002:**
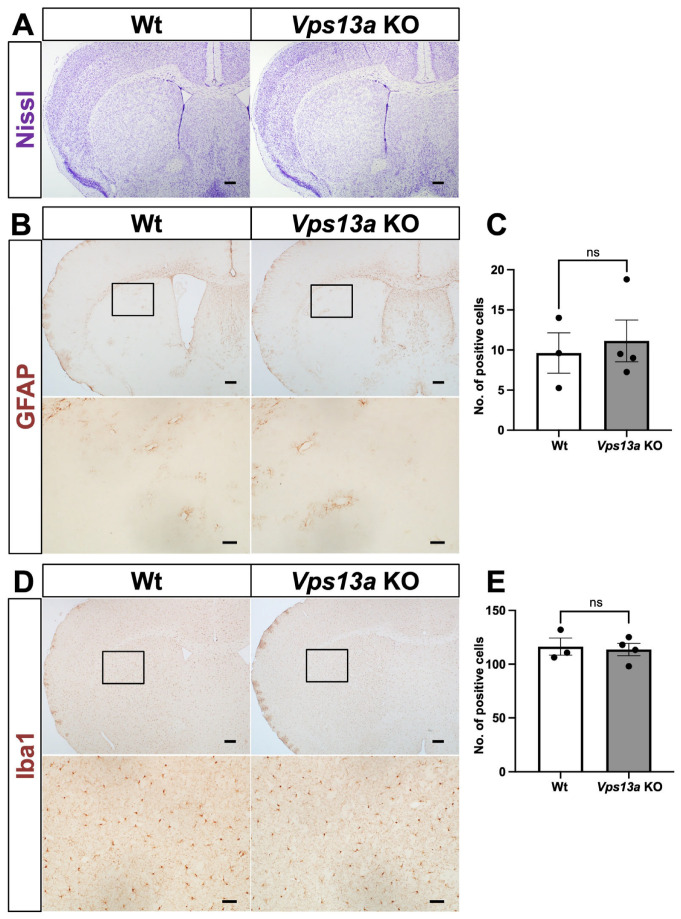
Brain phenotypes in the forebrain of *Vps13a* KO mice at 3 months old. (**A**) Nissl staining was performed on coronal sections of the forebrain from Wt (left) and *Vps13a* KO (right) mice (*n* = 3/genotype). Nissl staining allows for the visualization of neuronal cell bodies and overall tissue morphology. (**B**) GFAP immunohistochemistry (IHC) observing astrocyte activation. Coronal sections of Wt (left) and *Vps13a* KO (right) brains stained with anti-GFAP antibody (*n* = 3–4/genotype). The DAB staining (brown signals) indicates GFAP-positive astrocytes. (**C**) The number of GFAP-positive astrocytes was quantified by counting the GFAP-positive cells with a nucleus counter-stained by hematoxylin in the Wt and *Vps13a* KO brains. (**D**) Iba1 IHC. Coronal sections from Wt (left) and *Vps13a* KO (right) brains were stained for Iba1 to assess microglia (*n* = 3–4/genotype). DAB staining (brown signals) indicates Iba1-positive microglia. (**E**) Quantification of Iba1-positive cells in coronal sections of the forebrain from Wt and *Vps13a* KO mice. Scale bars: 200 µm (upper panels) and 50 µm (lower panels). The results are shown as the means ± SEM; ns = not significant.

**Figure 3 ijms-25-07776-f003:**
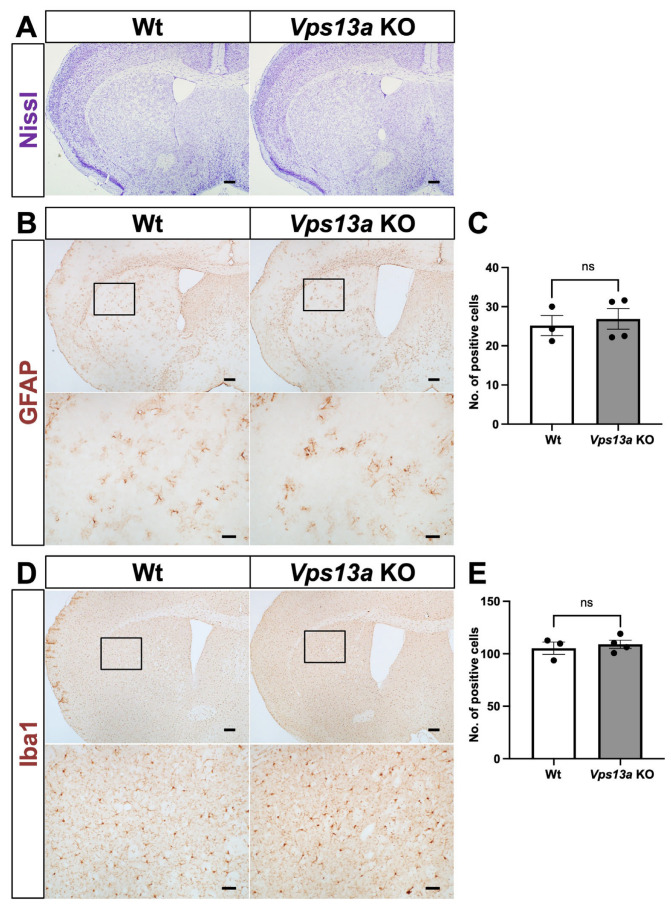
Brain phenotype in the forebrain of *Vps13a* KO mice at 18 months old. (**A**) Nissl staining was performed on coronal sections of the forebrain to observe the morphology of tissue from Wt (left) and *Vps13a* KO (right) mice at 18 months old (*n* = 3/genotype). (**B**) GFAP IHC was used to observe astrocyte activation. Coronal sections of Wt (left) and *Vps13a* KO (right) brains (*n* = 3–4/genotype) were stained with anti-GFAP antibody. The DAB staining indicates GFAP-positive astrocytes in the brown signals. (**C**) The numbers of GFAP-positive astrocytes were quantified by counting GFAP-positive cells with a nucleus counter-stained by hematoxylin in coronal sections from both Wt and *Vps13a* KO brains. (**D**) Iba1 IHC was used to assess microglia. Coronal sections from Wt (left) and *Vps13a* KO (right) brains (*n* = 3–4/genotype) were stained by anti-Iba1 antibody. The DAB staining indicates Iba1-positive microglia in the brown signals. (**E**) Quantification of Iba1-positive cells in coronal sections of the forebrain from Wt and *Vps13a* KO mice. Scale bars: 200 µm (upper panels) and 50 µm (lower panels). The results are shown as the means ± SEM; ns = not significant.

**Figure 4 ijms-25-07776-f004:**
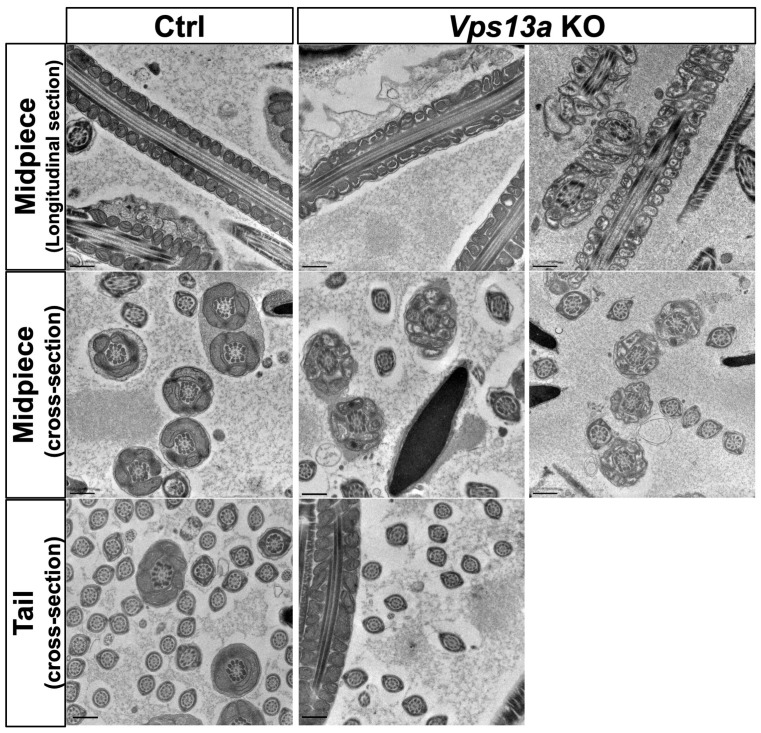
Transmission electron microscopy (TEM) analyses of sperm from *Vps13a* KO mice: (upper panels) TEM analyses of the longitudinal section of the midpiece region of sperm from Wt (left) and *Vps13a* KO mice (middle, right) at 16 weeks old (*n* = 2 animals in each genotype); (middle panels) TEM analyses of the cross-section of midpiece region of sperm from Wt (left) and *Vps13a* KO mice (middle, right) at 16 weeks old (*n* = 2 animals in each genotype); (lower panels) TEM analyses of the cross-section of the tail region of sperm from Wt (left) and *Vps13a* KO mice (middle, right) at 16 weeks old (*n* = 2 animals in each genotype). The 9 + 2 structures in the sperm flagellar axoneme were intact in the *Vps13a* KO sperm. The right and middle panels were taken from different *Vps13a* KO mice. Scale bars: 0.5 μm.

**Figure 5 ijms-25-07776-f005:**
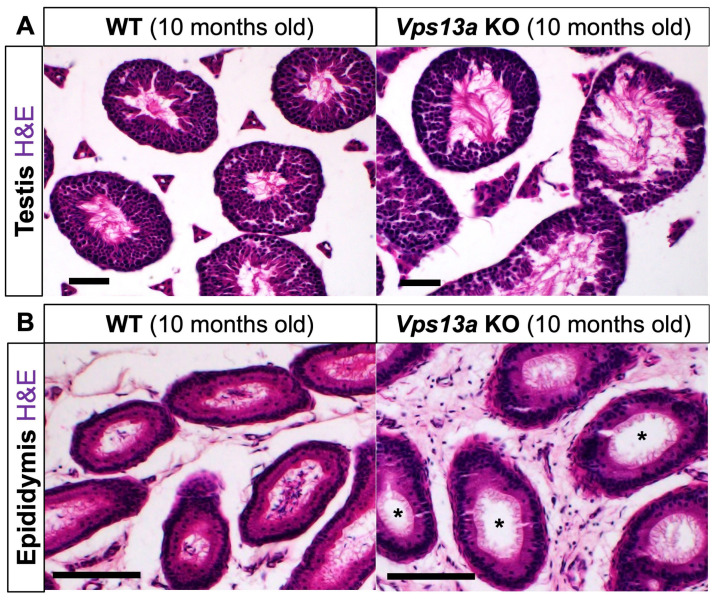
Histological analyses of testes: (**A**) hematoxylin and eosin (H&E) staining on testes sections of Wt (left) and *Vps13a* KO mice (right) at 10 months old (*n* = 3 animals in each genotype); (**B**) H&E staining on epididymis sections of Wt (left) and *Vps13a* KO mice (right) at 10 months old (*n* = 3 animals in each genotype). Asterisks indicate no sperm inside the epididymis. Scale bars: 50 μm.

**Figure 6 ijms-25-07776-f006:**
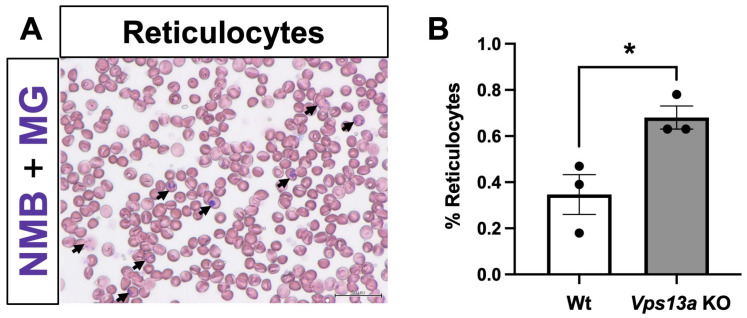
Peripheral blood smear tests: (**A**) reticulocytes among the red blood cells of the *Vps13a*-mutant mice are indicated by arrows, and New Methylene Blue (NMB) staining and May-Grunwald (MG) staining were performed; (**B**) quantification of the percentage of reticulocytes among the red blood cells of Wt and *Vps13a* KO mice at 18 months old (*n* = 3 animals in each genotype). Scale bars: 50 μm. The results are shown as the means ± SEM; * *p* < 0.05, unpaired *t*-test.

## Data Availability

The datasets used and studied during the current study are available from the corresponding author upon reasonable request.

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
