# Peer review of "Analysis of Brain, Blood, and Testis Phenotypes Lacking the *Vps13a* Gene in C57BL/6N Mice"

_ijms, 2024, doi:10.3390/ijms25147776_

Round 1
Reviewer 1 Report
Comments and Suggestions for Authors
In the present manuscript entitled “Analysis of brain, blood, and testis phenotypes lacking Vps13a gene in the C57BL/6N mice” the authors found that the Vps13a KO mice exhibited male infertility and abnormal sperm morphology. The conducted research is very interesting. This issue still has not been described in detail in the literature so far. The advantages of the manuscript: this is the first report of the Vps13a flox allele in B6N background; the figures of very good quality.
The manuscript is well written. However, below there are minor points, which should be taken into account.
1. The authors described that “The abnormal sperm morphology led us to examine male reproductive organs. H&E staining revealed dilated seminiferous tubules in the Vps13a KO testis at 10 months old (Figure 5A)” (page 5, line 144). Did the authors perform morphometric analyses, taking into account the diameter and area of the seminiferous tubules? Were the described changes observed in all tubules or only in some of them? Were only cross-sectional sections through the seminiferous tubules analyzed (omitting oblique or longitudinal sections that could distort the actual image of the changes)? Did the authors also observe other morphological disorders (e.g. inappropriate arrangement of germ cells in seminiferous epithelium, the presence of prematurely sloughed cells of the spermatogenic pathway into the lumen)?
2. The topic is very interesting, but the discussion is too short. It would be worth including a little more information in this regard.
3. Authors should add conclusions.
Author Response
Reviewer 1
We would like to thank the reviewer for constructive and critical comments.
In the present manuscript entitled “Analysis of brain, blood, and testis phenotypes lacking Vps13a gene in the C57BL/6N mice” the authors found that the Vps13a KO mice exhibited male infertility and abnormal sperm morphology. The conducted research is very interesting. This issue still has not been described in detail in the literature so far. The advantages of the manuscript: this is the first report of the Vps13a flox allele in B6N background; the figures of very good quality.
The manuscript is well written. However, below there are minor points, which should be taken into account.
Comment 1: The authors described that “The abnormal sperm morphology led us to examine male reproductive organs. H&E staining revealed dilated seminiferous tubules in the Vps13a KO testis at 10 months old (Figure 5A)” (page 5, line 144). Did the authors perform morphometric analyses, taking into account the diameter and area of the seminiferous tubules? Were the described changes observed in all tubules or only in some of them? Were only cross-sectional sections through the seminiferous tubules analyzed (omitting oblique or longitudinal sections that could distort the actual image of the changes)? Did the authors also observe other morphological disorders (e.g. inappropriate arrangement of germ cells in seminiferous epithelium, the presence of prematurely sloughed cells of the spermatogenic pathway into the lumen)?
Response1: Thank you very much for the critical and constructive comments. We performed morphometric analyses and added the data (lines 139-140). We analyzed cross-sections of the seminiferous tubules and omitted oblique or longitudinal sections that could distort the actual image of the changes. We also described the method, citing a paper in the materials and methods section (lines 324-335). As for other morphological disorders, it was difficult to provide quantitative data. Therefore, we only described the above-mentioned dilated lumens in seminiferous tubules of the Vps13a KO mice.
Comment 2: The topic is very interesting, but the discussion is too short. It would be worth including a little more information in this regard.
Response 2: Thank you very much for the critical comment. To improve the discussion section, we discussed testis and blood phenotypes to understand better the observed phenotypes (lines 202-209, 218-224).
Comment 3:. Authors should add conclusions.
Response 3: We added a conclusion section in the text (lines 324-335).
h for the comment. We modified the references according to the Journal style.

Reviewer 2 Report
Comments and Suggestions for Authors
In this study the Authors investigated the effects of knockout (KO) of vacuolar protein sorting 13A (VPS13A) gene in C57BL/6N mice on brain, blood and testis phenotypes. The Reviewer suggests that the following comments would be helpful to improve the quality of the manuscript.
Comments are as follows:
1. There is no clear defined objective of the study. The Authors presented the result of the study (L65-74) and should remove them, and give the objective(s).
2. In the M&M section should consider to provide all the appropriate references for the assays, for examples, Genotyping PCR, Western Blotting (What is the Bicinchoninic acid assay? Was the RIPA lysing buffer used?). This is important because a lot of fragments in this section have been re-copied by other sources according to the iThenicate report (for examples, L231-234/L241-249). Also, re-check the Abstract and Introduction to provide the appropriate references for fragments taken from other sources without the appropriate references.
3. The Discussion is incomplete and should consider to provide more detailed interpretations of the obtained results and concluding statements. How does Vps13a KO mice affect male infertility? Give other references to show that, besides the sperm parameters analyzed in this study, other measures of infertility have been evaluated in other studies.
4. Recheck the Journal style for the references given throughout the text.
Author Response
Reviewer2
We would like to thank the reviewer for constructive and critical comments.
In this study the Authors investigated the effects of knockout (KO) of vacuolar protein sorting 13A (VPS13A) gene in C57BL/6N mice on brain, blood and testis phenotypes. The Reviewer suggests that the following comments would be helpful to improve the quality of the manuscript.
Comments are as follows:
- There is no clear defined objective of the study. The Authors presented the result of the study (L65-74) and should remove them, and give the objective(s).
Thank you very much for the constructive comment. As suggested, we defined the study's objective and removed the sentences describing the results (lines 62-67).
- In the M&M sectionshould consider to provide all the appropriate references for the assays, for examples, Genotyping PCR, Western Blotting (What is the Bicinchoninic acid assay? Was the RIPA lysing buffer used?). This is important because a lot of fragments in this section have been re-copied by other sources according to the iThenicate report (for examples, L231-234/L241-249). Also, re-check the Abstractand Introduction to provide the appropriate references for fragments taken from other sources without the appropriate references.
Thank you very much for the comments. We cited appropriate references and re-wrote the Materials and Methods section. The bicinchoninic acid assay is used for protein quantification, and we used a kit from TaKaRa biotech company. It is also described (line 276).
- The Discussionis incomplete and should consider to provide more detailed interpretations of the obtained results and concluding statements. How does Vps13a KO mice affect male infertility? Give other references to show that, besides the sperm parameters analyzed in this study, other measures of infertility have been evaluated in other studies.
Thank you very much for the critical comment. We added descriptions of male infertility, citing a new reference (Arai et al., 2024), to improve the discussion (lines 202-209).
- Recheck the Journal style for the references given throughout the text.
Thank you very much for the comment. We modified the references according to the Journal style.
Round 2
Reviewer 2 Report
Comments and Suggestions for Authors
The Authors have satisfactorily addressed my comments, except for a few points that need to be addressed.
1. The objectives of the study are still unclear, and lack clarity. Remove the statement from L65-66 ("Our data show…")
Possible suggestion: "The main objective of this study was to generate a novel….. " Incomplete statement: "investigated phenotypes by histological analyses" Phenotypes of what? "brain, blood and testis …."
2. In the caption of Figure 2. Remove "*p < 0.05, unpaired t-test" and indicate the meaning of "ns". Do the same for Figure 3.
Author Response
We would like to thank the reviewer for constructive comments.
The Authors have satisfactorily addressed my comments, except for a few points that need to be addressed.
- The objectives of the study are still unclear, and lack clarity. Remove the statement from L65-66 ("Our data show…")
Possible suggestion: "The main objective of this study was to generate a novel….. " Incomplete statement: "investigated phenotypes by histological analyses" Phenotypes of what? "brain, blood and testis …."
We greatly appreciate your kind comments and suggestions. We removed the description of the results (L65-66). In addition, we described the objective of this study as suggested.
- In the caption of Figure 2. Remove "*p < 0.05, unpaired t-test" and indicate the meaning of "ns". Do the same for Figure 3.
Thank you very much for your kind suggestion. We removed the description and explained “ns” in the figure legends of Fig 2,3 and sFig1.